# Moving from drought hazard to impact forecasts

Samuel J. Sutanto [1]*, Melati van der Weert[1], Niko Wanders [2], Veit Blauhut[3] & Henny A.J. Van Lanen [1]

Present-day drought early warning systems provide the end-users information on the ongoing and forecasted drought hazard (e.g. river flow deficit). However, information on the forecasted drought impacts, which is a prerequisite for drought management, is still missing. Here we present the first study assessing the feasibility of forecasting drought impacts, using machine-learning to relate forecasted hydro-meteorological drought indices to reported drought impacts. Results show that models, which were built with more than 50 months of reported drought impacts, are able to forecast drought impacts a few months ahead. This study highlights the importance of drought impact databases for developing drought impact functions. Our findings recommend that institutions that provide operational drought early warnings should not only forecast drought hazard, but also impacts after developing an impact database.

[1] Hydrology and Quantitative Water Management Group, Environmental Sciences Department, Wageningen University and Research, Droevendaalsesteeg 3a, 6708PB Wageningen, The Netherlands. [2] Department of Physical Geography, Utrecht University, Princetonlaan 8A, 3508CB Utrecht, The Netherlands. [3] Hydrology Department, University of Freiburg, Fahnenbergplatz D-79098 Freiburg, Germany. *email: samuel.sutanto@wur.nl

Drought is one of the most damaging natural hazards in terms of affected people and economic cost[1–3]. Between 1900 and 2010, worldwide two billion people were affected and more than 10 million people passed away, because of the impacts of drought[3,4]. Such damages and losses are expected to become greater due to the projected increase of drought in multiple regions across the world under global warming[5,6]. To reduce drought impacts, actions have been undertaken including the development of drought monitoring and early warning systems[7–9]. Yet, present Drought Early Warning Systems (DEWSs) only produce early warning signal on the drought as a natural hazard, with encouraging skill for the first 3 months depending on the region and season[10–13]. These systems do not directly translate the drought hazard into the occurrence and severity of drought impacts. This missing tangible information, however, is crucial for water managers, stakeholders, and policy makers to better understand and prepare to drought[14].

Links between reported drought impacts and drought hazards, derived from standardized meteorological drought indices, e.g. Standardized Precipitation Index (SPI) and Standardized Precipitation Evaporation Index (SPEI), have been established in previous studies[15–17]. Here, we have conducted the first study that is moving one step forward from reconstructing historical drought impacts and associated risks[15,18] to forecasting drought impacts, up to a lead time (LT) of 7 months ahead. Here, first we developed drought impact forecasting functions for European administration regions (Nomenclature of Units for Territorial Statistics, NUTS-1)[19] using reported drought impacts obtained from the European Drought Impact Inventory from 1990 to 2017 (EDII[20], Supplementary Fig. 2), time series of drought hazard indicators from 1990 to 2017, and a Random Forest machine-learning algorithm[21]. Gridded meteorological observations of precipitation and evaporation were used to simulate runoff with a state-of-the-art hydrological model LISFLOOD[10,22]. These observations and simulations were used to define the historical drought hazard using the drought indices SPI[23], SPEI[24], and Standardized Runoff Index (SRI[25]), with different temporal aggregation periods (1, 3, 6, and 12 months). Finally, the developed drought impact functions were then utilized to translate the re-forecasted drought hazards through the set of drought indices from January 2002 to December 2010 into drought impact forecasts for certain sectors (impact categories) and each month with LTs of up to 7 months ahead and for 15 ensemble members (see Method section).

We show that the developed drought impact functions are capable of forecasting drought impacts months in advance with considerable skill up to 3–4 months, depending on the number of reported impacts, drought hazard severity, and drought duration. Models, that were trained using reported impacts for more than 50 months, generated high skill. The prediction has higher skill for longer drought events than short ones.

## Results

### Grouping the drought impacts
The credibility of drought impact functions developed with, for example, the Random Forest method largely depends on the completeness of the drought impact database. Model performance is therefore related to the quality and availability of its underlying data, as it is a well-known weakness of any data-driven approach[26]. By nature droughts are a rare phenomena, making it difficult to obtain enough drought impact data for a data-driven approach. Even though, we used the well-populated drought impact database, namely the EDII for Germany, large differences between the German NUTS-1 regions occur in frequencies and impact categories (Supplementary Fig. 1 and 2). For example, Bremen (HB) only reported water quality

impact while no other impact categories were reported. One should note that HB is a city-state that has no forest and agriculture fields. To address this shortcoming, we grouped the impacts into four groups to produce more robust drought impact forecast functions: Group 1 consists of agriculture and livestock farming, and forestry; Group 2 consists of energy and industry, water-borne transportation, and public water supply; Group 3 consists of water quality, freshwater ecosystem, and terrestrial ecosystem; and Group 4 consists of wildfire, air quality, and human health and public safety (see Supplementary notes).

### The skill of drought impact forecasts
Using the reported impacts for the grouped drought impact categories and the forecasted impacts obtained from the drought hazards derived from the re-forecasted drought indices, we calculated the performance of the developed drought impact functions trained using all data from 1990 to 2017 for each of the German NUTS-1 regions (Fig. 1). The performance of drought impact functions is indicated with the Relative Operating Characteristic (ROC) score[27] (see method section). For the shorter LTs, the drought impact functions show a reasonable skill with ROC values above 0.7 (greenish color). For the longer LTs, the models do not show any skill in most situations with ROC values below 0.5. Figure 1 confirms that the skill of drought impact forecasts resembles the skill of drought hazard forecasts, which is in general up to 3–4 months in advance[10–13]. In some regions, however, a certain drought impact can be predicted 7 months ahead, as shown by green colors, e.g. in Baden-Württemberg (BW) and Brandenburg (BB) for impact Group 2 and Group 3, respectively. This is especially the case for situations where prolonged drought impact periods were observed.

Figure 1 also shows for which impact groups and regions impacts can be forecasted, which is associated with the availability of reported impacts. In BW and Bavaria (BV), where they have more than 120 months and 150 months of reported impact, respectively, and have a wide variety of impact types (Supplementary Fig. 2), the drought impact function can be built for 3 impact groups, with skill up to 4 months. Other regions, where the reported impacts are less than 50 months, the drought impact function can be developed only for 2 impact groups, with skill up to 3 months. This indicates that the development of drought impact functions and its skill is region-impact group specific, and that it is strongly depended on the amount of reported impacts and types (total 18 out of 64).

### Re-forecasted likelihood of impact occurrence (LIO)
To illustrate the potential of the developed drought impact forecasting functions, we show examples of the re-forecasted likelihood of impact occurrence (LIO) modeled for each month and every LT for the German NUTS-1 region Rheinland-Pfalz (RP) and Brandenburg (BB) for drought impact Groups 2 and 3, respectively, which are closely linked to hydrological drought (Fig. 2). Figure 2 shows that the drought impact forecast functions have evoked impact signals in 2003 and 2006. The EDII, however, does not report impacts for Group 2 in 2006 for RP and LT = 1 month (Fig. 2a). This can be either a false alarm or an absence of reported drought impacts. Prediction based on the median ensemble for LT = 3 months does not show a signal anymore for drought impact in 2006 for RP. The amplitude of the drought impact signal for 2003 has also been decreased, but it is still visible. On the other hand, drought impacts in Group 3 in 2003 and 2006 for BB can be predicted well with the strong signal up to 5 months in advance (Fig. 2b). Overall, signals become weaker with increasing lead times and the uncertainty of the predictions becomes larger, denoted by the wider red-shaded area around the

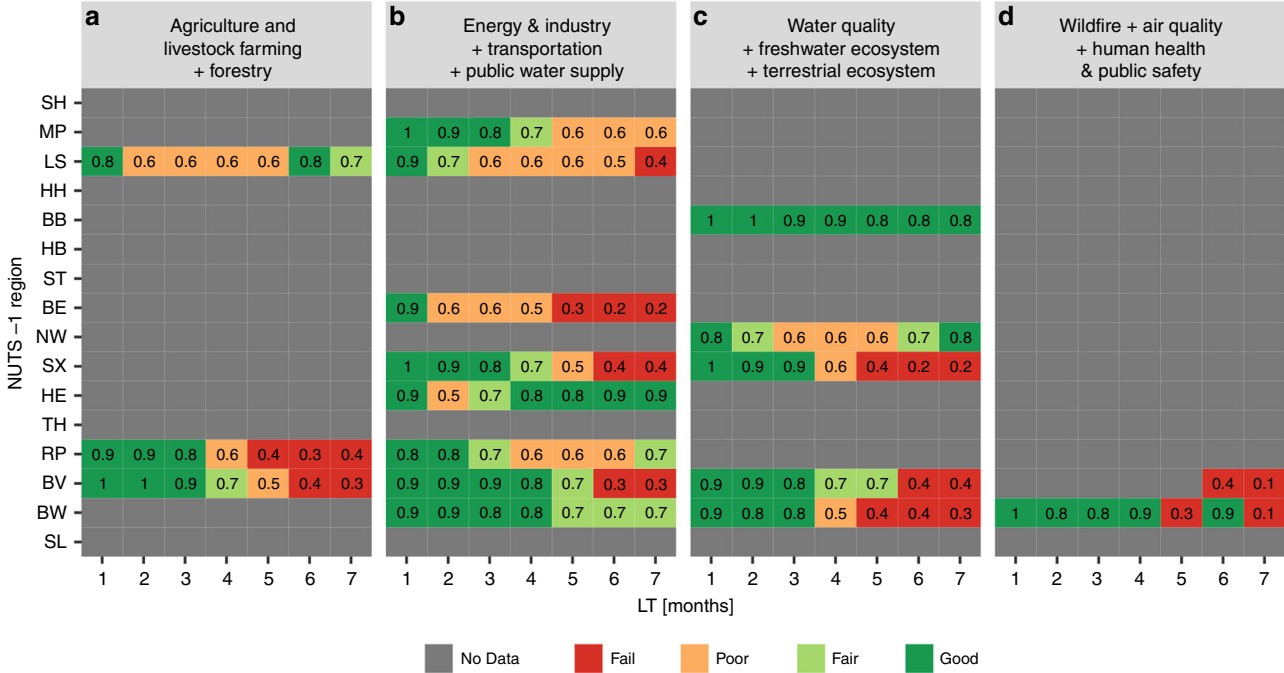

**Fig. 1** ROC skill scores of re-forecasted drought impacts (2002–2010) on 4 groups of impact categories against reported impacts. NUTS-1 regions (see Supplementary Fig. 1) are given on the y-axis and the LT on the x-axis. Red, orange, light green, and green colors indicate the performance of drought impact functions, which are fail, poor, fair, and good, respectively, following ref. [61]

median. This likely is related to the larger uncertainty in drought hazard forecasting beyond 3 months[11–13].

Drought in 2003 and 2006 are well-known drought events in Europe because of the impacts and its associated heatwaves[28–30]. To check whether drought occurred in RP in 2006, we plotted the drought events from 2000 to 2010 using SPI-6 and SRI-6 to approach hydrological drought (see Supplementary Fig. 3 and Supplementary notes). Our analysis using SPI-6 indicates that severe droughts occurred at RP and BB in 2003, 2006, and 2008. Supplementary Fig. 3 shows that the drought in 2006 likely has triggered impacts in both regions. The 2006 drought in RP had a high severity and long duration. Moreover, drought has occurred in those regions since 2003 and precipitation events in 2005 might not have completely alleviated the hydrological drought (e.g., drought in streamflow and groundwater). Therefore, we anticipate that likely impact occurred, but was not reported for RP in 2006 (Fig. 2a). Long drought duration from 2003 to 2006 produced a stronger and more robust hydrological drought impact signal in 2006 for BB.

Figure 3a presents an example of the median forecasted LIO using re-forecast data for each month with LT = 1–7 months ahead from January 2002 to December 2010 for the BB region and drought impact Group 3 (blue lines). This figure illustrates the result of drought impact forecasting as if it would be encapsulated in a drought EWS. For no-drought conditions, the forecast functions produce low LIO values. LIO values increase before drought events indicating that drought impact can be predicted a few months before. We also notice some high LIO values but no drought was reported, i.e. the drought year 2008 and 2010[28,31], which could trigger unneeded preparation measures. However, there is a clear difference between high LIOs that led to drought impacts in 2003 and 2006, with high LIOs that did not result in no drought impact (2008 and 2010). In 2003 and 2006, the LIO values increase from LT ≥ 1 months, and later these gradually decrease after drought impact occurred. For 2008, the drought impact forecast functions only produce rather low and

short-lived LIOs, whereas, for 2010, the functions generate a high LIO for LT = 1, but it sharply decreases for LT > 1. When interpreting the forecasted drought impact signal, this exhibits first, that an increasing LIO from LT = 1 to LT > 1 is more important than a decreasing LIO. Second, a sharp decreasing LIO indicates a low probability of an impact occurrence. The duration of the drought events in 2008 and 2010 might also have been too short to create a noticeable impact in this region, hence it causes a steep decrease of the LIO for LT > 1.

We find that the duration of reported impacts plays an important role in how many months ahead drought impacts can be forecasted. For reported short-lived impacts, e.g. the 2003-drought, the drought impact Group 3 for the BB region could be forecasted 2 months Prior To (PT) the observed impact (PT = 2, Fig. 3b). For long-lived impacts, e.g. the 2006 drought, however, the drought impacts that were reported in March 2006, were already forecasted starting from 7 months ahead (September 2005, PT = 7, red color Fig. 3c). For this particular drought event, all forecasts done at different months prior to the start of the reported impact show a similar pattern of increasing LIOs, which started with increased LIOs from January 2006 onwards. Drought impact forecasts for the BB region demonstrate that drought impact can be forecasted 2 months before for shorter-lived reported impacts.

## Discussion
A subset of the EDII[20] was used to build statistical relationships between drought hazards using drought indices and its reported impacts, as a basis for drought impact forecasts. There are some uncertainties in the impact dataset, which affect the overall performance of developed drought impact functions. Previous studies have shown temporal and geographical biases in the EDII inventory (Supplementary Fig. 2)[20,32]. Moreover, reported drought impact information used in this study may not cover all drought events and impacts that actually happened, for every

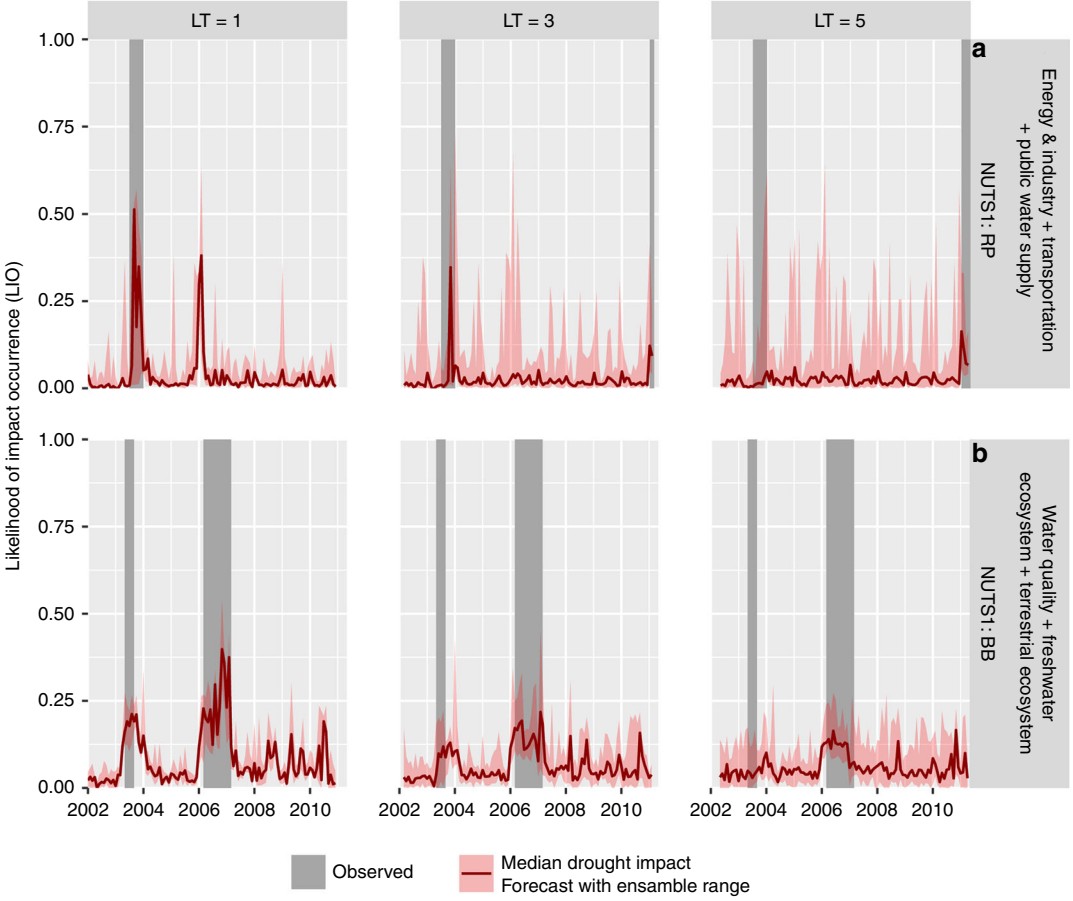

**Fig. 2** Examples of the re-forecasted Likelihood of Impact Occurrence (LIO). The figures show the examples of the re-forecasted LIO for three selected lead times (LT = 1, 3, and 5 months) for selected German NUTS-1 regions, i.e. Rheinland-Pfalz (RP) for impact Group 2 (**a**), and Brandenburg (BB) for impact Group 3 (**b**). The red line indicates the median re-forecasted LIO. The red-shaded area around the median indicates the ranges in which all ensemble members fall. The gray bars indicate periods with reported impacts (EDII[20])

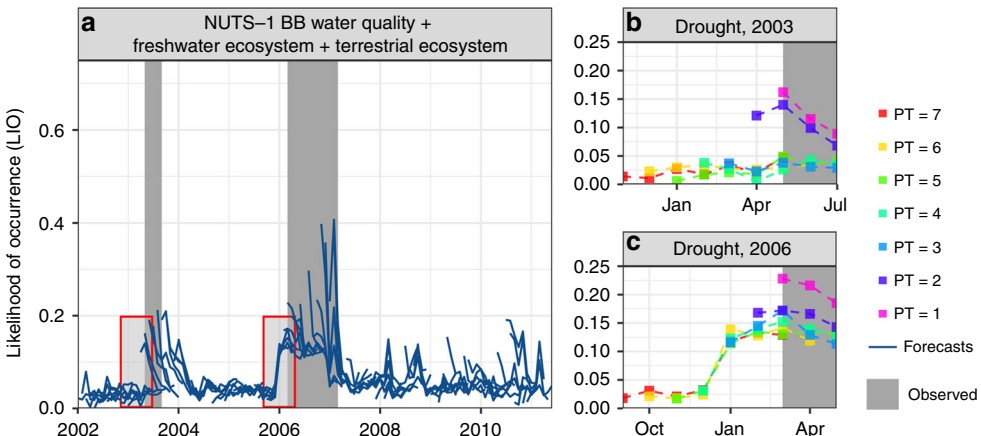

**Fig. 3** Illustration of the median re-forecasted LIO obtained from a drought early warning system. **a** Example of the re-forecasted LIO and reported drought impact (EDII, gray bars) for a German NUTS-1 region Brandenburg (BB) for the drought impact Group 3. Blue lines are the re-forecasted median LIO done for each month from January 2002 to December 2010 with LT = 1–7 months. **b** Detailed overview of the 2003 drought until July (impact forecasts start in November 2002), indicated by a red box in (**a**). **c** Detailed overview of the 2006 drought until May (impact forecasts start in September 2005), indicated by a red box in (**a**). The various colored lines give the forecasted LIO done at different times prior to (PT) the observed drought impact period for different LTs

region. The lack of impact report in the database can either means the absence of a drought impact or that a drought impact was not reported in the EDII. Both possibilities can differ regionally due to e.g. reporting behavior, affected sector or research interest of past studies. The forecasted drought impact for the RP region in 2006 might be an example of no reported impact (Fig. 2a) although the drought indices (e.g., SPI-6 and SRI-6) shows long drought duration at this region (Supplementary Fig. 3a and b, respectively). We suggest to consider the vulnerabilities and exposure of the impacts in each studied region, which can provide a good measure for drought impact forecasting[33].

We developed drought impact forecast functions including all available hydro-meteorological drought indices with different temporal accumulation periods (SPI-x, SPEI-x, and SRI-x, with x = 1, 3, 6, 12 months). The Random Forest algorithm was simply used as a classifier without preceding predictor selection. This means that we did not a priori exclude short accumulation periods of meteorological drought indices (i.e., SPI-1 or SPEI-1) in the Random Forest model as important predictors for hydrological type of impacts, such as energy and industry, water-borne transportation, and public water supply (Group 2) (also seen in ref. [34]). In our case, the best performing drought predictors are region and impact group specific (see Supplementary Fig. 4 and Supplementary notes). Our findings show that in general, SPEI and SRI are better drought impact predictors than the SPI. Links between accumulation periods of drought indices with certain drought impacts suggest that short accumulation periods are best associated with the meteorological type of drought impacts and vice versa for the hydrological type of drought impacts[35,36]. Therefore, the Random Forest algorithm seems capable to distinguish these links, although it is a black box model. We used all accumulation periods to develop the drought impact forecasting functions. A priori selection of a number of accumulation periods for a certain type of drought group might result into improved impact forecasts, e.g. accumulation periods of 1 and 3 months for agriculture, and accumulation periods of 6 and 12 months for water-borne transportation, and public water supply.

We acknowledge that the performance of the drought impact forecast functions depends on the completeness of its underlying reported impact data. Besides Germany, drought impact forecast functions may also be developed for other European regions, which have sufficient drought impact data, such as the UK (1855 reports), the Netherlands (219 reports), and France (525 reports)[20,34]. The European Drought Observatory[7] and the ANYWHERE Drought EWS are encouraged to develop and later encapsulate the drought impact forecast functions for these regions into their drought early warning systems since our study shows that drought impacts can be forecasted a few months in advance. Outside Europe, the US drought portal established by Climate Prediction Center (NOAA)[37] has the potential to develop the drought impact functions by utilizing drought impact information from the Drought Impact Reporter (DIR), launched by the National Drought Mitigation Center (NDMC)[38]. In China, drought impact forecast functions for agriculture impact could be developed by using a seasonal hydrological forecasting system that has been developed by ref. [39] and the agricultural drought impact database collected by the China Meteorological Administration[40]. However, this can only be done, if it appears that sufficient drought impact data are available. Therefore, we highlight the importance of an ample drought impact database for developing drought impact functions. We encourage stakeholders and water managers in each country to collect and collaborate in providing comprehensive information on drought impacts.

Attempts to seasonal drought impact forecasting using dynamic climate models in combination with deterministic impact models (top-down approach) are already known for some time for particular purposes, e.g. agricultural crop yields[41–44] or navigation on main German rivers[45]. Efforts are limited to specific crops, regions or rivers. Likely, a combination of a bottom-approach, as described in this study, which has weaknesses in the sense of completeness of impact reporting, with top-down approaches is the best way forward on drought impact forecasting. This is also advocated in climate change adaptation studies (e.g., ref. [46]). The top-down part of the combined approach for drought impact forecasting could be broadened by implementing experiences with integrated environmental modeling that are being used for a wide range of scenario analyses[47], including water resources systems[48,49].

This study is pioneering in linking reported drought impacts to drought hazard forecasts. Drought impact forecast functions were developed to predict the drought impacts with a lead-time of up to 7 months ahead. Results from regions in Germany were used to illustrate the forecasting. Our study demonstrates the importance of comprehensive and complete drought impact databases and the applicability of drought impact functions to forecast drought impacts a few months in advance, depending on drought impact duration. Therefore, we highly recommend the institutions that provide operational drought early warning systems to encourage developing a drought impact database and later to move one step forward to provide drought hazard prediction as well as drought impact forecasts. The information of predicted drought impacts can be of great value for water managers, stakeholders, and policy makers in understanding and planning drought management responses in due time.

## Methods

**Data**. A flowchart showing the data and methods used in this study is presented in Supplementary Fig. 5. Three datasets were used in this study to develop the drought impact forecasting functions: first, gridded meteorological data (box a, Supplementary Fig. 5) and runoff data simulated by the LISFLOOD model fed by observations (hereafter referred as proxy observed data, box d), second, re-forecast hydro-meteorological datasets (box b and e), and third, reported drought impacts taken from the European Drought Impact Inventory database (EDII, box c).

Information on drought impacts was extracted from the EDII database[20]. This is a database where thousands of reports on drought impacts (~10,000 reports) are compiled from 33 European countries, which started as an initiative of the EU FP-7 project Drought RSPI. The information on drought impacts at German state level from the EDII inventory was then transformed into a monthly binary time series of impact and no impact for each impact category (box j). Please see ref. [20] for detailed information on the EDII database.

A proxy of observed hydro-meteorological data was used to calculate drought indices to represent the drought hazard. The proxy observed data (from 1990 to 2017) as well as re-forecast data (from 2002 to 2010), were provided by the European Centre for Medium-Range Weather Forecasts (ECMWF) as part of the European Flood Awareness System (EFAS)[22,50]. In the EFAS, meteorological observation data are collected from ground observations (>5000 synoptic stations), obtained from various sources, such as the Global Telecommunication System of the WMO, the Joint Research Center (JRC) meteorological database, and high-resolution data received from the National member States institutions[51]. These meteorological data are precipitation, potential (reference) evapotranspiration rate (PET) using the Penman-Monteith, potential evaporation rate from open water and bare soil, and temperature. The re-forecasts of hydro-meteorological variables have a lead-time of 215 days (circa 7 months) and consist of 15 ensemble members[52]. For this study, we used the re-forecast data run with ECMWF SEAS4. Ref. [10] provides detailed information on proxy observed and re-forecasts data taken from EFAS.

**Standardized drought indices**. Hydro-meteorological standardized drought indices were used to identify the severity of the drought. These indices indicate the degree of dryness by providing the deviation from the long-term mean, i.e. number of standard deviations. In this study, we used the Standardized Precipitation Index (SPI[23]), the Standardized Precipitation Evaporation Index (SPEI[24]), and the Standardized Runoff Index (SRI[25]) (box f). The SPI-x was calculated by fitting a probabilistic distribution on monthly precipitation data, whereas the SPEI-x was calculated by fitting a probabilistic distribution on the climatic water balance (monthly precipitation minus potential evapotranspiration). The SRI-x was calculated by fitting a probabilistic distribution on monthly runoff. In the present study, SPI-x, SPEI-x, and SRI-x were calculated for the following accumulation periods: x = 1, 3, 6, and 12 months. The calculation of the standardized indices

transforms the monthly hydro-meteorological data into 12 distributions for each index, accumulation period, and for every month of the year. SPI and SRI were calculated using the gamma distribution that can be described by two parameters: $\alpha$ (the shape parameter) and $\beta$ (the inverse scale parameter)[23,25], whereas SPEI was calculated using the three-parameter log-logistic distributed variables $\alpha$, $\beta$, and $\lambda$ (origin parameters)[24] (box g). The gamma distribution has quite a flexible shape parameter, which is applicable to the wide range of accumulated precipitation in EU[53,54]. These distribution parameters then were used to calculate the observed drought events (box h) and re-forecast drought events (box i). We used the same data and method as in ref. [55] to calculate observed and forecasted drought events using the standardized indices.

**Drought impact function derived from random forest**. Random Forest (RF) is a Machine Learning algorithm that is based on classification and regression trees[21] and it is a powerful algorithm to develop a predictive model. Some studies have utilized RF to monitor and predict meteorological drought hazard based on historical data[56–58]. References [16,34] are the first that employed RF to link drought indices and reports from the EDII. These studies, however, have not used RF for the drought impact forecasting. We selected RF amongst many other machine learning algorithms because RF produces randomly numerous independent tress as an ensemble to avoid overfitting and sensitivity to training data configuration (in our case 2000 trees), the predictive performance of RF has similar performance as the best-supervised learning algorithms, RF efficiently estimates the test error without incurring the cost of repeated model training associated with cross-validation, RF is flexible and has very high accuracy, and last but not least RF has been widely used for drought studies and produces better performance compared to other Machine Learning approached (e.g., Boosted regression trees, cubist, decision trees, Hurdle, and logistic regression[16,56,58]).

The binary time series of the standardized drought indices (predictor variable) together with the reported drought impacts (response variable) were used to translate drought hazards into the likelihood of drought impact occurrences using the RF algorithm[21] (box k). A random forest model was created for every German NUTS-1 region and a specific drought impact group. The Random Forest creates a multitude of randomly uncorrelated decision trees. Each tree is constructed based on a bootstrapped subsample of the data. Predictions are made by the mean prediction of the individual trees. The Random Forest method was performed using the Random Forest package in R studio[59]. There was no prior predictor selection in the Random Forest algorithm and therefore we used SPI-x, SPEI-x, SRI-x, month, and year as the predictors. To identify the best set of drought indicators linked to the impact occurrences, the Caret feature was used[60]. This feature uses the prediction accuracy on the out-of-bag portion for both the full models and after permuting each predictor variable. The differences between the two models were then averaged over all trees, and normalized by the standard error. We assessed the descriptive power of the model using observational data from 1990 to 2017. For this analysis, the models were trained on a subset of observed data from 1990 to 2017. The observed data from 1990 to 2015 were used for testing purposes and the validation was carried out using data from 2016 to 2017 and the EDII reports. In our manuscript, however, we only presented the drought impact forecasts from 2002 to 2010 (reforecast data) simulated using the models that were trained using historical observed data from 1990 to 2017. Likelihood of Impact Occurrence (LIO) was estimated by calculating the probability of the number of tress (Ni) that indicated impact. For each NUTS-1 region in Germany and for each impact group, drought impact functions were derived using the Random Forest analysis based on the standardized drought indices using proxy hydro-meteorological data and drought impact time series (box l). The Random Forest algorithm used to build drought impact forecasting functions is explained in more detail in refs. [16,34]. These functions were then used to predict drought impacts using re-forecasted drought indices (box m).

**The drought impact forecasting skill score**. The skill of the drought impact forecasts was evaluated by comparing the re-forecasted drought impact against the observed drought impacts taken from the EDII database for the period 2002 to 2010 (box n). Time series of the observed impacts and re-forecasted drought impacts were therefore translated into binary time series of impact or no impact occurrence for the calculation of drought impact forecasting skill score. The skill/performance of the developed drought impact functions to forecast drought impacts was assessed using a commonly used method called the Relative Operating Characteristic (ROC)[27]. This skill score has been used among many studies dealing with probabilistic forecasts[11,13]. The ROC curve was used as a criterion to measure the discriminate ability between two outcomes. The ROC curve gives the relation between the true positive rate (sensitivity) and the false positive rate (specificity). The Area Under the Curve (AUC) was calculated to measure the accuracy of the forecast. The larger the area, the more accurate the forecast will be. The AUC has a range from [1,0] where 1 is a perfect score. All values beneath $AUC = 0.5$ indicate no skill. A color-coding was applied to divide the forecast skill into 4 categories. Red color stands for fail (ROC < 0.5), orange color stands for poor performance ($0.5 \leq ROC < 0.7$), light green color stands for fair performance ($0.7 \leq ROC < 0.8$), and green color stands for good performance ($ROC \geq 0.8$)[61].

## Data availability

The historical forcing data and seasonal re-forecasts from EFAS are accessible through the MARS archive at ECMWF (https://apps.ecmwf.int/mars-catalogue/?class=c3) under a COPERNICUS open data license. Other data generated and/or analyzed during this study are available from the corresponding author on request (S.J.S.).

## Code availability

All codes used to conduct the analysis presented in this paper can be obtained by contacting the corresponding author (S.J.S.).

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

## Acknowledgements

The research is supported by the ANYWHERE project (Grant Agreement No. 700099), which is funded within EU's Horizon 2020 research and innovation program http://www.anywhere-h2020.eu. The hydro-meteorological output came from the EFAS computational center, which is part of the Copernicus Emergency Management Service (EMS) and Early Warning Systems (EWS) funded by framework contract number 198702 of the European Commission. Finally, we acknowledge the Ministry of Science, Research and the Arts of the State of Baden-Württemberg for financing the DRIeR-project, which maintains EDII. This research is part of the Wageningen Institute for Environment and Climate Research (WIMEK-SENSE) and it supports the work of UNESCO EURO FRIEND-Water program and the IAHS Panta Rhei project Drought in the Anthropocene.

## Author contributions

All authors conceived and implemented the research. Data analyses, model output analyses, and all figures have been performed by M.V.D.W. S.J.S. and H.A.J.V.L. wrote the initial version of the paper. N.W. and V.B. contributed to interpreting the results, discussion, and improving the paper.

## Competing interests

The authors declare no competing interests.
