## [Peer Review File · Nature Communications]

Reviewers' comments:

Reviewer #2 (Remarks to the Author):

The Authors present a study “assessing the feasibility of forecasting drought impacts, using machine-learning to relate forecasted hydro-meteorological drought indices to reported drought impacts”.

As requested by the Editor, I will only provide my opinion on the machine learning technique aspect of the manuscript.

Machine learning techniques have already been frequently used in drought prediction problems. The following is a list of articles that could be interesting for the Authors:

- Belayneh, A., & Adamowski, J. (2012). Standard precipitation index drought forecasting using neural networks, wavelet neural networks, and support vector regression. *Applied computational intelligence and soft computing*, 2012, 6.
- Belayneh, A., Adamowski, J., Khalil, B., & Ozga-Zielinski, B. (2014). Long-term SPI drought forecasting in the Awash River Basin in Ethiopia using wavelet neural network and wavelet support vector regression models. *Journal of Hydrology*, 508, 418-429.
- Belayneh, A., Adamowski, J., Khalil, B., & Quilty, J. (2016). Coupling machine learning methods with wavelet transforms and the bootstrap and boosting ensemble approaches for drought prediction. *Atmospheric research*, 172, 37-47.
- Deo, R. C., & Şahin, M. (2015). Application of the extreme learning machine algorithm for the prediction of monthly Effective Drought Index in eastern Australia. *Atmospheric Research*, 153, 512-525.
- Deo, R. C., Kisi, O., & Singh, V. P. (2017). Drought forecasting in eastern Australia using multivariate adaptive regression spline, least square support vector machine and M5Tree model. *Atmospheric Research*, 184, 149-175.
- Deo, R. C., Tiwari, M. K., Adamowski, J. F., & Quilty, J. M. (2017). Forecasting effective drought index using a wavelet extreme learning machine (W-ELM) model. *Stochastic environmental research and risk assessment*, 31(5), 1211-1240.
- Park, S., Im, J., Jang, E., & Rhee, J. (2016). Drought assessment and monitoring through blending of multi-sensor indices using machine learning approaches for different climate regions. *Agricultural and forest meteorology*, 216, 157-169.
- Rhee, J., & Im, J. (2017). Meteorological drought forecasting for ungauged areas based on machine learning: Using long-range climate forecast and remote sensing data. *Agricultural and Forest Meteorology*, 237, 105-122.

- Roodposhti, M. S., Safarrad, T., & Shahabi, H. (2017). Drought sensitivity mapping using two one-class support vector machine algorithms. *Atmospheric research*, 193, 73-82.

Random Forest is a very powerful algorithm for developing predictive models. However, it is not effective for all possible forecast problems, of course. The Authors should clarify some aspects related to the use of the algorithm:

- Why Random Forest algorithm is applied in this study? What are the other feasible alternatives? What are the advantages of adopting this particular machine learning technique over others in this case? How will this affect the results? More details should be provided.
- How are the input vectors of the model made up? Has the sensitivity to the variables been investigated? More details should be provided.
- Which cross validation technique was used for the development of the model?
- What is the size of the training dataset? And that of testing and validating datasets?
- What is the number of trees in the forest? Which stopping rule was adopted?
- What criteria were used to assess the accuracy of the model? What results did they provide?

Based on the above considerations, I believe that the article needs significant improvements concerning the description of the forecasting model based on the Random Forest algorithm. The effectiveness of the model in this specific case should also be more clearly demonstrated.

Reviewer #3 (Remarks to the Author):

Summary

This paper uses reported drought impacts to develop a relationship between drought hazard indicators which is then used to forecast drought impacts. It used observed meteorological and modelled runoff data to develop the relationship between indicators (SPI, SPEI and SRI) and observed drought impacts (from the EDII) using Random Forest models. These relationships were then used to 'forecast' impacts using ECMWF hindcast data from 2002 to 2010.

This paper takes the concept of understanding how drought indicators are related or can be linked to drought impacts which provides more context to drought indicators which may be used in a drought monitoring system, a step further in order to forecast impacts. It represents a novel addition to the literature; and the content, motivation and impact are of high quality. I'm sure this paper will be the first of many which utilise the EDII and other impact datasets to forecast drought

impacts in Europe and elsewhere in the world. It represents a step forwards in the ability to plan and prepare for droughts more effectively, so mitigating impacts on society and the environment.

As elaborated below, the paper needs some clarifications and improved English in places, but I suggest this paper is accepted after minor revisions.

Decision: Minor revisions

Overall comments

The English and grammar was in general good, however, there were a few points where perhaps not quite the right word was used. I have noted these in the manuscript mark up (and in places suggested alternatives).

I disagree that impact forecasting should replace hazard forecasting, as in some scenarios and for some applications know the rainfall/river flow deficits etc. that are likely to occur will be of interest – for example for water supply managers dependent on river flow abstractions. This could be a factor of the language used but this point should be modified to say that impact forecasts complement the hazard forecasts that are more commonly produced which will certainly still be of use to some water users, supply chains and commercial users of forecast information.

In the methods you mention that the SRI is derived using LISFLOOD outputs, but then it is not really discussed in the paper (nor shown in plots etc.). You state on page 2 that “we plotted the drought events from 2000 to 2010 using SPI-6 to approach hydrological drought” Would it not be more appropriate to use the SRI here? Second, is the SPI-6 the most appropriate accumulation of SPI to represent hydrological drought? Van Loon & Lahaa (2015; <https://doi.org/10.1016/j.jhydrol.2014.10.059>) and Barker et al. (2016; <https://doi.org/10.5194/hess-20-2483-2016>) found that SPI accumulation periods best related to hydrological droughts varied depending on catchment properties. What is the reasoning for selecting SPI-6 here?

It would be useful to add a comment on the choice of distribution used, including some reference to papers that have tested the distributions used for deriving standardised drought indicators such as Stagge et al. (2015, <https://doi.org/10.1002/joc.4267>) where they tested the best distribution for SPI and SPEI across Europe. The choice of distribution can affect the indicator values given which can have important implications on the declaration of droughts (e.g. Nunez et al. 2014, <https://doi.org/10.1016/j.jhydrol.2014.05.038>) but will also affect the relationship between indicators and impacts and therefore the forecasting of impacts.

The Likelihood of Impact Occurrence (LIO) as the main way of presenting the results in Fig 2 and 3, yet it is not mentioned in the methods – please add some text on how this was calculated and references, for example Blauhut et al. 2015 (Environ. Res. Lett. 10 014008).

Thinking about how this impact forecasting method would work running in real time, I miss a comment on the possibility of calculating SPEI in real time. Also, what method you use to derive PET for the SPEI calculation, for example the data needed for Penman Monteith are not available in real time, and temperature based PET may not have much inter-annual variation resulting in only small differences between SPI and SPEI.

Some notes on the figures:

- Fig 3, the acronym PT is somewhat confusing. In 3a, the blue lines represent the forecasts at different lead times, 3b and 3c are zoomed in sections of 3a but the lines (which from what I can see are zoomed in on the red boxes in 3a) are now labelled as different times prior to the impact, not lead times. Does PT not equal LT? I think it would be best to be consistent here or at least introduce the acronym PT earlier in the caption that it is currently (i.e. at the end) – perhaps just before describing 3b.
- It would be best to have what is currently Fig S2 as Fig S1, so the NUTS acronyms are introduced before they are used in (what is currently) S1.
- In Fig S4, please label the y axis of the bar charts (e.g. predictor importance, or importance).
- Fig S5 - Box k, l and m could be moved below the level of box h and i to make it visually clearer these steps occur after all other steps have been completed (it will also avoid the arrows crossing which would be clearer too).

Reply to reviewers

We would like to thank the reviewer for the valuable comments and suggestions. In this document, we reply to each of the comments.

(PxLaa-bb: Px refers to page number x, and Laa-bb to line numbers aa to bb).

Reviewer 2		
No	Comment	Reply
1	Machine learning techniques have already been frequently used in drought prediction problems. The following is a list of articles that could be interesting for the Authors (Ref. 1-9).	We will refer to some of the previous studies on drought prediction using Machine Learning (ML) in our revised manuscript that the reviewer suggested. However, we want to emphasize that none of them has used ML for the drought impact forecasting. To authors' knowledge, our manuscript is the first that studies drought impact forecasting (P1L20-23).
2	Random Forest is a very powerful algorithm for developing predictive models. However, it is not effective for all possible forecast problems, of course. The Authors should clarify some aspects related to the use of the algorithm:	We do agree with the reviewer that Random Forest (RF) is a powerful algorithm to develop a predictive model (P4L196-197). Hence, we believe RF is well suited to develop a predictive model that links drought hazard indices that are rather easily to forecast, to drought impacts. Below, we clarify the reasons.
	- Why Random Forest algorithm is applied in this study? What are the other feasible alternatives? What are the advantages of adopting this particular machine learning technique over others in this case? How will this affect the results? More details should be provided.	Reasons that RF was chosen are: a) RF produces randomly numerous independent trees as an ensemble to reduce the chance of overfitting and reduces the sensitivity to the selected split sample training data configuration. b) RFs are often used in many geophysical applications, making them familiar to the final user community that can use these impact-based forecasts. c) The predictive performance of RF is similar to the best-supervised learning algorithms. d) RF efficiently estimates the test error without incurring the effort of repeated model training associated with cross-validation. e) RF is flexible and has very high accuracy. f) RF has been widely used for drought prediction studies and produces better performance compared to other ML approaches (e.g., Boosted regression trees, cubist, decision trees, Hurdle, and logistic regression; Park et al., 2016; Rhee and Im, 2017; Bachmair et al., 2017). g) RF has been successfully employed in Europe to link drought indices and the drought impact database (Bachmair et al., 2016; Bachmair et al., 2017) for developing drought impacts functions, though without using these for forecasting. Based on these reasons, we decided to use

		the RF method. Beside RF, Hurdle and Log regression models have also been used for developing drought impact functions, showing inferior results than RF (Blauhut et al., 2015; Stagge et al., 2015; Blauhut et al., 2016). Please note that those studies reconstructed historical conditions and were not used for drought impact forecasting, which is the novelty of our paper (P4L197-P5L206).
	- How are the input vectors of the model made up? Has the sensitivity to the variables been investigated? More details should be provided.	For the input, we used multiple time series of drought indices as predictor variables and for the response variable we used a binary time series, which consisted of impact or no impact for each month derived from the EDII database (P5L207-209). To analyze the sensitivity of the model, we used the Caret feature, which uses the prediction accuracy on the out-of-bag (OOB) portion for both the full model and after permuting each predictor variable (P5L213-216). We presented the output as predictor importance (Fig. S4).
	- Which cross validation technique was used for the development of the model?	We did not do cross validation. However, we did OOB performance analysis for the development of our RF model, which is not exactly the same but has connections with cross validation (CV) (P5L214-217). We think that the calculation of OOB error in the model training phase is sufficient to test the performance of the model (also see point 2c).
	- What is the size of the training dataset? And that of testing and validating datasets?	We assessed the descriptive power of the model using observational data from 1990-2017. For this analysis, the models were trained on a subset of observed data from 1990 to 2017. The observed data from 1990-2015 were used for testing purposes and the validation was carried out using data from 2016 to 2017 and the EDII reports. In our manuscript, however, we only presented the drought impact forecasts from 2002-2010 (reforecast data) simulated using the models that were trained using historical observed data from 1990 to 2017 (P5L216-220).
	- What is the number of trees in the forest? Which stopping rule was adopted?	The number of trees that we used was 2000 trees. We did not apply a stopping rule (P5L202).
	- What criteria were used to assess the accuracy of the model? What results did they provide?	The performance of the forecasted drought indices and impacts were assessed using commonly used methods, e.g., Relative Operating Characteristic Curve (ROC curve). This method has been used by many studies dealing with probabilistic forecasts to evaluate the skill of the forecasts. The ROC curve gives the relation between the true positive rate (sensitivity) and the false positive rate (specificity). The Area Under the

		Curve (AUC) was calculated to measure the accuracy of the forecast. The larger the area, the more accurate the forecast will be. The AUC has a range from [1,0] where 1 indicates a perfect forecast. All values beneath the diagonal line (AUC=0.5) indicate no skill (P5L226-238) .
3	Based on the above considerations, I believe that the article needs significant improvements concerning the description of the forecasting model based on the Random Forest algorithm. The effectiveness of the model in this specific case should also be more clearly demonstrated.	We thank the reviewer for his/her suggestions, which helped strengthening the manuscript in its current form. Additional information and clarification about the RF method were added to the revised manuscript in the Method section, although we already stated in the manuscript that detailed information about RF could be obtained from the references (Bachmair et al., 2016, 2017). We do not think that we should duplicate too much (P4-P5, drought impact function derived from RF) .
Reviewer 3		
No	Comment	Reply
1	This paper takes the concept of understanding how drought indicators are related or can be linked to drought impacts which provides more context to drought indicators which may be used in a drought monitoring system, a step further in order to forecast impacts. It represents a novel addition to the literature; and the content, motivation and impact are of high quality. I'm sure this paper will be the first of many which utilize the EDII and other impact datasets to forecast drought impacts in Europe and elsewhere in the world. It represents a step forwards in the ability to plan and prepare for droughts more effectively, so mitigating impacts on society and the environment.	We would like to thank the reviewer for the acknowledgement of the novelty of our paper.
2	I disagree that impact forecasting should replace hazard forecasting, as in some scenarios and for some applications know the rainfall/river flow deficits etc. that are likely to occur will be of interest – for example for water supply managers dependent on river flow abstractions. This could be a factor of the language used but this point should be modified to say that impact forecasts complement the hazard forecasts that are more commonly produced which will certainly still be of use to some water users, supply chains and commercial users of forecast information.	We do agree with the reviewer. It was not our intention to suggest that forecasting drought hazards should be replaced by forecasting drought impacts. We suggest that institutions that provide drought hazard forecasts, may consider to move one step forward by forecasting drought impacts as well. We changed the sentence according to the suggestion (P4L156-158) .
3	In the methods you mention that the SRI is derived using LISFLOOD outputs, but then it is not really discussed in the paper (nor shown in plots etc.). You state on page 2 that “we plotted the drought events from 2000 to 2010 using SPI-6 to approach hydrological drought” Would it not be more	The use of SRI was discussed in paragraph 2 in the Discussion section, as well as the SRI was plotted in Figure S4 (P3L118-123) . We had chosen the SPI index in Figure S3 because SPI is more widely used than SRI. This figure was included just to show that there were droughts in 2003 and 2006 in

	appropriate to use the SRI here? Second, is the SPI-6 the most appropriate accumulation of SPI to represent hydrological drought? Van Loon & Lahaa (2015) and Barker et al. (2016) found that SPI accumulation periods best related to hydrological droughts varied depending on catchment properties. What is the reasoning for selecting SPI-6 here?	Germany. We added SRI-6 in Figure S3 (P14). The optimal accumulation period for standardized drought indices depends on catchment characteristics (e.g. fast versus slowly-responding catchments), but also on the impacted sector. For some sectors, which largely depend on soil moisture, an accumulation period of 3 months (SPI-3) fits well, for other sectors that are more influenced by groundwater, or groundwater-fed rivers, longer accumulation periods are selected (e.g. SPI-6). For instance, the heat maps compiled by Bloomfield et al. (2013) show that accumulation periods over 6 months (SPI-x, x>6) are typical for groundwater. We have modified the manuscript accordingly (P9L411-416).
4	It would be useful to add a comment on the choice of distribution used, including some reference to papers that have tested the distributions used for deriving standardised drought indicators such as Stagge et al. (2015) where they tested the best distribution for SPI and SPEI across Europe. The choice of distribution can affect the indicator values given which can have important implications on the declaration of droughts (e.g. Nunez et al. 2014) but will also affect the relationship between indicators and impacts and therefore the forecasting of impacts.	As suggested by the reviewer, we have add the reasoning behind the selection of probability distributions to the manuscript (P4L192-193).
5	The Likelihood of Impact Occurrence (LIO) as the main way of presenting the results in Fig 2 and 3, yet it is not mentioned in the methods – please add some text on how this was calculated and references, for example Blauhut et al. 2015.	LIO in our study using RF was estimated by calculating the probability of the number of trees (Ni) that indicated impact. The explanation on how the LIO was calculated was added in the revised manuscript (P5L221-222).
6	Thinking about how this impact forecasting method would work running in real time, I miss a comment on the possibility of calculating SPEI in real time. Also, what method you use to derive PET for the SPEI calculation, for example the data needed for Penman Monteith are not available in real time, and temperature based PET may not have much inter-annual variation resulting in only small differences between SPI and SPEI.	We agree with the reviewer that it is important to hypothesize how the impact based forecast system will work in a real-time operational setting. Most importantly, we can take advantage of the fact that our impact forecasts depend on the monthly standardized indices timeseries (SPI, SPEI, and SRI). The advantage is that only a minimal amount of information is added in the final days, as such latency in the data delivery will only have a minor impact on the final monthly values. Even in extreme cases with a latency in the data of 7 days, this would only contain <5% of the daily data that goes into the indices calculation (7 days out of a total 215). Nonetheless we want to reduce this latency as much as possible which is why we take advantage of some developments and opportunities mentioned below.

		Firstly, All hydro-meteorological data used in this study are obtained from the European Flood Alert System (EFAS), run by ECMWF. In the EFAS, meteorological observation data are collected from ground observations (>5000 synoptic stations), obtained from various sources, such as the Global Telecommunication System of the WMO, the Joint Research Center (JRC) meteorological database, and high-resolution data received from the National member States institutions (Pappenberger et al., 2011). These meteorological data are precipitation, potential (reference) evapotranspiration rate (PET), potential evaporation rate from open water and bare soil, and temperature. For PET, the Penman-Monteith method is used (which has been added to the revised manuscript P4L173-178). Secondly, for real time monitoring, EFAS runs the observed meteorological data up to -18 hours prior to start of the hydrological forecast simulations. To fill the gap, a short 18-hour LISFLOOD simulation is run, driven by either DWD (German meteorological forecasts) or ECMWF deterministic forecasts. Finally, we can take advantage of the fact that seasonal forecast are only produced once a month. The data is released in the first week of the month (around day 5) after all the forcing data were run up to the end of the previous month. For seasonal forecasts hence we use the gridded observed meteorological data (including Penman-Monteith PET) and no forecast data were added. These data are used for the calculation of the SPI and SPEI.
7	Fig 3, the acronym PT is somewhat confusing. In 3a, the blue lines represent the forecasts at different lead times, 3b and 3c are zoomed in sections of 3a but the lines (which from what I can see are zoomed in on the red boxes in 3a) are now labelled as different times prior to the impact, not lead times. Does PT not equal LT? I think it would be best to be consistent here or at least introduce the acronym PT earlier in the caption that it is currently (i.e. at the end) – perhaps just before describing 3b.	We used different acronyms here to avoid misunderstanding between LT and PT. Lead time (LT) is the time since the forecast was issue. For example for the forecast done in January, LT = 2 months means month February. While the PT describes the time period in months prior to the observed impact. We introduced the acronym earlier in the revised manuscript to avoid confusion for the readers (P3L100-103).
8	It would be best to have what is currently Fig S2 as Fig S1, so the NUTS acronyms are introduced before they are used in (what is currently) S1.	We shifted the Figures accordingly (P13-14).
9	In Fig S4, please label the y axis of the bar charts (e.g. predictor importance, or importance).	The label was added for the grey histogram plots (P15).

10	Fig S5 - Box k, l and m could be moved below the level of box h and i to make it visually clearer these steps occur after all other steps have been completed (it will also avoid the arrows crossing which would be clearer too).	The box k, l, m, and n were moved below h and i (P16).
11	Please see the pdf mark up for specific comments.	Thanks for all your suggestions. We elaborated your suggestions in the revised manuscript.

References

1. Belayneh, A., & Adamowski, J. (2012). Standard precipitation index drought forecasting using neural networks, wavelet neural networks, and support vector regression. *Applied computational intelligence and soft computing*, 2012.
2. Belayneh, A., Adamowski, J., Khalil, B., & Ozga-Zielinski, B. (2014). Long-term SPI drought forecasting in the Awash River Basin in Ethiopia using wavelet neural network and wavelet support vector regression models. *Journal of Hydrology*, 508, 418-429.
3. Belayneh, A., Adamowski, J., Khalil, B., & Quilty, J. (2016). Coupling machine learning methods with wavelet transforms and the bootstrap and boosting ensemble approaches for drought prediction. *Atmospheric research*, 172, 37-47.
4. Deo, R. C., & Şahin, M. (2015). Application of the extreme learning machine algorithm for the prediction of monthly Effective Drought Index in eastern Australia. *Atmospheric Research*, 153, 512-525.
5. Deo, R. C., Kisi, O., & Singh, V. P. (2017). Drought forecasting in eastern Australia using multivariate adaptive regression spline, least square support vector machine and M5Tree model. *Atmospheric Research*, 184, 149-175.
6. Deo, R. C., Tiwari, M. K., Adamowski, J. F., & Quilty, J. M. (2017). Forecasting effective drought index using a wavelet extreme learning machine (W-ELM) model. *Stochastic environmental research and risk assessment*, 31(5), 1211-1240.
7. Park, S., Im, J., Jang, E., & Rhee, J. (2016). Drought assessment and monitoring through blending of multi-sensor indices using machine learning approaches for different climate regions. *Agricultural and forest meteorology*, 216, 157-169.
8. Rhee, J., & Im, J. (2017). Meteorological drought forecasting for ungauged areas based on machine learning: Using long-range climate forecast and remote sensing data. *Agricultural and Forest Meteorology*, 237, 105-122.
9. Roodposhti, M. S., Safarrad, T., & Shahabi, H. (2017). Drought sensitivity mapping using two one-class support vector machine algorithms. *Atmospheric research*, 193, 73-82.
10. Van Loon & Lahaa (2015; <https://doi.org/10.1016/j.jhydrol.2014.10.059>).
11. Barker et al. (2016; <https://doi.org/10.5194/hess-20-2483-2016>).
12. Stagge et al. (2015, <https://doi.org/10.1002/joc.4267>).
13. Nunez et al. (2014, <https://doi.org/10.1016/j.jhydrol.2014.05.038>).
14. Blauhut et al. (2015, *Environ. Res. Lett.* 10 014008).
15. Bachmair, S., Svensson, C., Prosdocimi, I., Hannaford, J. & Stahl, K. Developing drought impact functions for drought risk management. *Nat. Hazards and Earth Syst. Sci.* 17, 1947–1960 (2017).
16. Bachmair, S., Svensson, C., Hannaford, J., Barker, L. J. & Stahl, K. A quantitative analysis to objectively appraise drought indicators and model drought impacts. *Hydrol. Earth Syst. Sci.* 20, 2589-2609 (2016).
17. Stagge, J.H., Kohn, I., Tallaksen, L.M. and Stahl, K. (2015). Modeling drought impact occurrence based on meteorological drought indices in Europe. *J. Hydrol.*, 530, 37-50, <http://dx.doi.org/10.1016/j.jhydrol.2015.09.039>.
18. Blauhut, V., Stahl, K., Stagge, J.H., Tallaksen, L.M., De Stefano, L. and Vogt, J. Estimating risk across Europe from reported drought impacts, drought indices, and vulnerability factors. *Hydrol. Earth Syst. Sci.* 20, 2779-2800 (2016).
19. Pappenberger, F., Thilen, J. and Del Medico, M. The impact of weather forecast improvements on large scale hydrology: analysis a decade of forecasts of the European Flood Alert System. *Hydrol. Process.* 25, 1091-1113 (2011).

20. Bloomfield, J. P., & Marchant, B. P. (2013). Analysis of groundwater drought building on the standardized precipitation index approach. *Hydrol. Earth Syst. Sci.*, 17, 4769–4787.

REVIEWERS' COMMENTS:

Reviewer #2 (Remarks to the Author):

The authors satisfactorily addressed my comments. I have no further comments.

Reviewer #3 (Remarks to the Author):

The paper is much improved in clarity with changes made in the revisions. I have noted a few small items which could be changed to improve the language and clarity of the final paper. Well done, I think this is a great addition to the literature.

Reply to reviewer 3

We would like to thank the reviewer for the carefully reading and valuable suggestions. In this document, each of the comments is discussed.

(PxLaa: Px refers to page number x, and Laa to line numbers aa).

Reviewer 3		
No	Comment	Reply
1	P1L10: I think there's still some ambiguity here, so suggest these changes (or something similar) here, in addition to the position on P4 stated in the rebuttal.	We revised the last sentence of the Abstract to make it consistent with the last paragraph of the Discussion (P10).
2	P1L32: i.e. impact. This would clarify what is meant by categories here	We changed the phrasing to impact categories (P1L32).
3	P1L34	We changed the words accordingly (P1L34).
4	P1L36	We changed the word accordingly (P1L36).
5	P1L37: What is meant here? Good skill? Low uncertainty? Both?	Robust here means has higher skill. We revised the text accordingly (P1L37).
6	P2L44: Was just one water quality impact recorded in HB? There is either some word(s) missing here, or a plural missing.	Bremen only reported water quality impact. There is no other impact reported in Bremen. See Figure S2 (only orange color to be seen). We slightly revised the sentence to make more clear (P2L45).
7	P2L74	We switched the words accordingly (P2L74).
8	P2L78: the English here could be improved	We have rewritten the sentence in the revised manuscript (P2L78).
9	P3L108: using? via is not the right choice of word here	We changed the word accordingly (P3L108).
10	P3L134: Do you mean here that these projects are now encouraged to use the impact forecasts (because they can see that they work)? Or that you suggest they use them because the results are good? Please clarify as I think these are two different things.	We mean that these projects are encouraged to build and later encapsulate the drought impact functions into their drought EWSs since we showed in our study (i.e. study case Germany) that drought impacts can be forecasted a few months in advance. We rewrote the sentence for clarification (P3L134).
11	P3L135	We changed the text accordingly (P3L136).
12	P4L199: I think it would be best to say the EDII here for consistency/clarity	We changed the word accordingly (P4L199).
13	P10: spelling mistake 'Human health'	The typo was corrected (P10).
14	P15: spelling mistake 'Human health' and spelling mistake - High	The typos were corrected (P15).